# SAMPLE-BASED POINT CLOUD DECODER NETWORKS

## ABSTRACT

Point clouds are a flexible and ubiquitous way to represent 3D objects with arbitrary resolution and precision. Previous work has shown that adapting encoder networks to match the semantics of their input point clouds can significantly improve their effectiveness over naive feedforward alternatives. However, the vast majority of work on point-cloud decoders are still based on fully-connected networks that map shape representations to a fixed number of output points. In this work, we investigate decoder architectures that more closely match the semantics of variable sized point clouds. Specifically, we study sample-based point-cloud decoders that map a shape representation to a point feature distribution, allowing an arbitrary number of sampled features to be transformed into individual output points. We develop three sample-based decoder architectures and compare their performance to each other and show their improved effectiveness over feedforward architectures. In addition, we investigate the learned distributions to gain insight into the output transformation. Our work is available as an extensible software platform to reproduce these results and serve as a baseline for future work.

## 1 INTRODUCTION

Point clouds are an important data type for deep learning algorithms to support. They are commonly used to represent point samples of some underlying object. More generally, the points may be extended beyond 3D space to capture additional information about multi-sets of individual objects from some class. The key distinction between point clouds and the more typical tensor data types is that the information content is invariant to the ordering of points. This implies that the spatial relationships among points is not explicitly captured via the indexing structure of inputs and outputs. Thus, standard convolutional architectures, which leverage such indexing structure to support spatial generalization, are not directly applicable.

A common approach to processing point clouds with deep networks is voxelization, where point clouds are represented by one or more occupancy-grid tensors (Zhou & Tuzel (2018), Wu et al. (2018)). The grids encode the spatial dimensions of the points in the tensor indexing structure, which allows for the direct application of convolutional architectures. This voxelization approach, however, is not appropriate in many use cases. In particular, the size of the voxelized representation depends on the spatial extent of the point cloud relative to the spatial resolution needed to make the necessary spatial distinctions (such as distinguishing between different objects in LIDAR data). In many cases, the required resolution will be unknown or result in enormous tensors, which can go beyond the practical space and time constraints of an application. This motivates the goal of developing architectures that support processing point cloud data directly, so that processing scales with the number of points rather than the required size of an occupancy grid.

One naive approach, which scales linearly in the size of the point cloud, is to 'flatten' the point cloud into an arbitrarily ordered list. The list can then be directly processed by standard convolutional or fully-connected (MLP) architectures directly. This approach, however, has at least two problems. First, the indexing order in the list carries no meaningful information, while the networks do not encode this as a prior. Thus, the networks must learn to generalize in a way that is invariant to ordering, which can be data inefficient. Second, in some applications, it is useful for point clouds to consist of varying numbers of points, while still representing the same underlying objects. However, the number of points that can be consumed by the naive feedforward architecture is fixed.

PointNet (Qi et al., 2017) and Deepsets Zaheer et al. (2017) exhibit better performance over the MLP baseline with a smaller network by independently transforming each point into a high-dimensional representation with a single shared MLP that is identically applied to each individual point. This set of derived point features is then mapped to a single, fixed-sized dense shape representation using a symmetric reduction function. As such the architectures naturally scale to any number of input points and order invariance is built in as an architectural bias. As a result, these architectures have been shown to yield significant advantages in applications in which point clouds are used as input, such as shape classification.

The success of PointNet and DeepSet style architectures in this domain shows that designing a network architecture to match the semantics of a point cloud results in a more efficient, and better performing network. Since point clouds are such a useful object representation, it's natural to ask how we should design networks to decode point clouds from some provided shape representation. This would allow for the construction of point cloud auto-encoders, which could serve a number of applications, such as anomaly detection and noise smoothing. Surprisingly, the dominant approach to designing such a differentiable point cloud decoder is to feed the dense representation of the desired object through a single feedforward MLP whose result is then reshaped into the appropriate size for the desired point cloud. This approach has similar issues as the flat MLP approach to encoding point clouds; the decoder can only produce a fixed-sized point cloud while point clouds are capable of representing objects at low or high levels of detail; the decoder only learns a single deterministic mapping from a shape representation to a point cloud while we know that point clouds are inherently random samples of the underlying object.

The primary goal and contribution of this paper is to study how to apply the same lessons learned from the PointNet encoder's semantic congruence with point clouds to a point cloud decoder design. As such, we build on PointNet's principles to present the 'NoiseLearn' algorithm– a novel, simple, and effective point cloud decoding approach. The simplicity of the decoding architectures and the increase in performance are strong indicators that sample-based decoders should be considered as a default in future studies and systems. In addition, we investigate the operation of the decoders to gain insight into how the output point clouds are generated from a latent shape representation.

## 2    RELATED WORK

Point cloud decoders are a relatively unexplored area of research. Among the works which describe an algorithm that produces a point cloud, the majority focus their efforts on learning a useful latent shape representation that is then passed to a MLP decoder.

PU-Net (Yu et al., 2018) is one such example, in which they design a novel point cloud upsampling network which uses a hierarchical approach to aggregating and expanding point features into a meaningful latent shape representation. To decode the learned shape representation into a point cloud, the latent vector is then passed through a feedforward MLP to produce a fixed number of points. This implies that the network would need to be retrained to allow for a different upsampling rate, which unlikely to be a desired property of an upsampling algorithm.

TopNet (Tchapmi et al., 2019) recognizes the data inefficiency of using a single MLP to decode a point cloud and instead reorganizes their MLP into a hierarchical tree structure in which MLPs at the same level share the same parameters. Their results show that addressing this inefficiency allows for better performance with a smaller parameter count. Similarly, in "Learning Localized Generative Models for 3D Point Clouds via Graph Convolution" Valsesia et al. (2019) augments their decoder by assuming a graph structure over the decoded point cloud and employing graph convolutions. However, despite improved performance neither approach addresses the other issues that come with using MLPs to decode entire point clouds, namely the fixed-size output.

"Point Cloud GAN" (Li et al., 2018) and PointFlow (Yang et al., 2019) take a different approach to producing a point set in a generative setting. Instead of learning a single mapping from any latent vector directly to its decoded point cloud, they learn a function parameterized by the latent vector which transforms low-dimensional Gaussian noise to a 3D point on the surface of the object described by the latent shape representation. This sampling based approach is more in line with the semantics of point clouds. First, an arbitrary number of points can be drawn from the Gaussian noise to produce a point cloud consisting of that number of points without requiring any changes

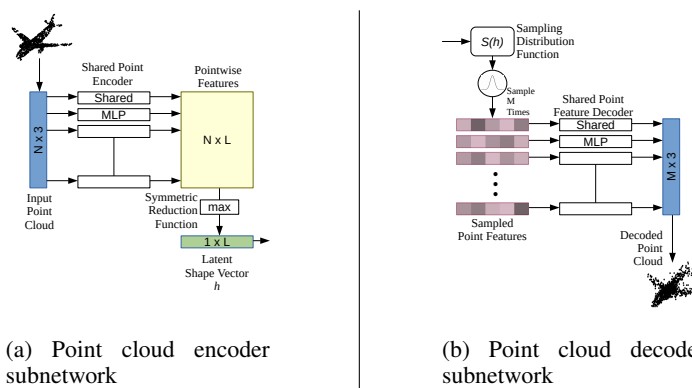

(a) Point cloud encoder subnetwork

(b) Point cloud decoder subnetwork

Figure 1: Diagrams demonstrating the operation of the PointNet style encoder network shared by all architectures we evaluate, and the parallel decoder network shared by all the sampling approaches.

to or retraining of the algorithm. Second, every individual point is decoded independently and identically, which avoids the data inefficiency issues that come with using MLPs to process set data.

While this sampling approach has several desirable properties and appears promising, it's unclear whether the algorithm is applicable outside of the GAN settings these two papers inhabit, if they require specific bespoke loss functions to be trained effectively, or if they are capable of outperforming the baseline MLP approach according to other metrics.

## 3 POINT-CLOUD DECODING ARCHITECTURES

A *point cloud* is a set of $n$ 3D points $\mathbb{C} = \{\boldsymbol{p}_1, \ldots, \boldsymbol{p}_N\}$, where each $\boldsymbol{p}_i \in \mathbb{R}^3$. In general, each $\boldsymbol{p}_i$ may have additional auxiliary information associated with it via non-spatial dimensions. While all of our architectures easily generalize to include such information, in this paper, we focus on point clouds that exclusively encode shapes without auxiliary information.

A *point cloud auto-encoder* takes a point cloud $\mathbb{C}$ with $n$ points and outputs a point cloud $\hat{\mathbb{C}}$ with $m$ points that is intended to represent the shape described by $\mathbb{C}$. While often $n = m$, we are interested in the general case when $n$ and $m$ may be different, which corresponds to up-scaling or down-scaling $\mathbb{C}$. Each auto-encoder will be comprised of an encoder $E(\mathbb{C})$, which takes an input point cloud and outputs a latent shape representation $\boldsymbol{h}$ in $\mathbb{R}^l$, and a decoder $D(\boldsymbol{h})$ which maps a latent representation to an output point cloud of the appropriate size. Thus, given an input point cloud $\mathbb{C}$, the auto-encoder output is given by $\hat{\mathbb{C}} = D(E(\mathbb{C}))$.

In this paper, we focus on the Chamfer distance as the measure of auto-encoder quality. Intuitively this loss function measures how well $\hat{\mathbb{C}}$ matches $\mathbb{C}$ in terms of the nearest neighbor in $\hat{\mathbb{C}}$ to each point in $\mathbb{C}$ and vice versa. Specifically, if $dist(\boldsymbol{p}, \hat{\mathbb{C}})$ gives the distance between point $\boldsymbol{p}$ and the nearest neighbor in point cloud $\hat{\mathbb{C}}$, our loss function is defined by $L(\mathbb{C}, \hat{\mathbb{C}}) = \frac{1}{n} \sum_{\boldsymbol{p} \in \mathbb{C}} dist(\boldsymbol{p}, \hat{\mathbb{C}}) + \frac{1}{m} \sum_{\boldsymbol{p} \in \hat{\mathbb{C}}} dist(\boldsymbol{p}, \mathbb{C})$.

Since the focus of this paper is on point-cloud decoders, all of our architectures use the same point-cloud encoder architecture, while varying the decoder architecture. Below, we first overview the common PointNet-style encoder used followed by a description of the four decoders considered in our experimental analysis, which include three sample-based decoders.

### 3.1 POINTNET ENCODER ARCHITECTURE

PointNet (Qi et al., 2017) handles unordered input by recognizing that a symmetric function $g$ (such as element-wise `max` or `sum`) produces the same result regardless of the order of its inputs. PointNet thus learns a single transformation function $f$ that maps individual points to an $l$-dimensional representation and then combines those representations via $g$. That is, the latent encoding produced by PointNet for a point cloud $\mathbb{C} = \{\boldsymbol{p}_1, \ldots, \boldsymbol{p}_n\}$ is the $l$ dimensional vector

$E(\mathbb{C}) = g\left(f(\boldsymbol{p}_1), \ldots f(\boldsymbol{p}_N)\right)$. As desired, $E(\mathbb{C})$ is invariant to the ordering of points in $\mathbb{C}$ and applies to any number of points.

We learn an MLP representation of $f$, with input space $\mathbb{R}^3$, encoding points, and output space $\mathbb{R}^l$, encoding the latent representation or *point feature*. We use max as the reduction function $g$ to map the arbitrary number of resulting point features to a single fixed-size latent shape representation. The hidden layers and size of the latent shape representation for each instantiation of this encoder architecture can be found in Table 1.

## 3.2 SAMPLE-BASED DECODERS

Most prior work has used *MLP decoders*, which we consider here as a baseline approach. An MLP decoder is a fully connected network that takes the latent shape representation as input and outputs an $m \times 3$ output vector, which represents the $m$ output points. Accordingly, MLP decoders are parameterized by the number and size of their fully connected layers. In our experiments, each fully connected layer consists of parameterized ReLU units with a batch normalization layer.

Our main focus is on sample-based decoders, which allow for an arbitrary number of outputs points to be produced for a latent shape representation. In particular, given a latent shape representation $\boldsymbol{h}$, each of our decoders is defined in terms of a point feature sampling distribution $S(\boldsymbol{h})$, where the decoder produces a point-cloud output by sampling $m$ point features from $S(\boldsymbol{h})$.

Once we have a set of $M$ independently sampled point features from our sampling distribution $S(\boldsymbol{h})$ we need to transform each one into a triple representing that point's location. Note that we are now in an identical but opposite situation as the point cloud encoder. Whereas the encoder had to transform independent point samples of some underlying object into corresponding high-dimensional representations, our decoder now has to transform independently sampled high-dimensional point representations into a point in space on the surface of the target object. Therefore, we can simply apply the same style of PointNet encoding mechanism with different input and output tensor sizes to implement an effective point feature decoder. The sizes of the hidden layers in our decoder network can be seen in Table 1. By applying the shared MLP point decoder to each sampled point feature, we can directly decode point clouds of arbitrary size.

Below we describe three architectures for $S$, which are compared to each other and the baseline MLP decoder in our experiments.

**NoiseAppend Decoder.** NoiseAppend is similar to the sampling approach described in "Point Cloud GAN" by Li et al. (2018). They sample point features by simply sampling from a multivariate Gaussian distribution with zero mean and unit variance before appending the sampled noise to the latent shape vector. That is, $S(\boldsymbol{h}) = \texttt{concat}\left(\boldsymbol{h}, \mathcal{N}(0, \boldsymbol{I})\right)$.

However, this requires us to decide how many elements of noise should be appended to the latent shape representation. Li et al. (2018) state that the size of the appended noise vector should be 'much smaller than' the size of the latent shape representation, but it's not clear how much noise is necessary to allow the decoder to fully represent the shape. Ultimately this is an additional hyperparameter that needs to be investigated and tuned.

**NoiseAdd Decoder.** NoiseAdd builds on the concept of adding unit Gaussian noise to the latent shape vector with the goal of avoiding the additional hyperparameter that NoiseAppend introduces. This can be easily accomplished by treating the entire latent vector as the mean of a Gaussian distribution with unit variance. That is, $S(\boldsymbol{h}) = \mathcal{N}(\boldsymbol{h}, \boldsymbol{I})$.

However, this violates the claim by Li et al. (2018) that the amount of noise introduced to the resulting point feature samples should be much smaller than the size of the latent shape representation itself. Therefore, it may be the case that uniformly adding noise to every element of the latent vector obscures the crucial information it represents.

**NoiseLearn Decoder.** NoiseLearn attempts to instead learn a small separate function $V(h)$ which predicts the log-variance of the desired point feature distribution. Specifically, $S(\boldsymbol{h}) = \mathcal{N}\left(\boldsymbol{h}, e^{V(\boldsymbol{h})/2}\boldsymbol{I}\right)$. We define $V(\boldsymbol{h})$ as a small MLP, the size of which can be seen in Table 1.

By allowing the network to choose the amount and location of noise to be added to the latent shape vector, we hope that it will learn both to add an appropriate amount of noise for the target shape while

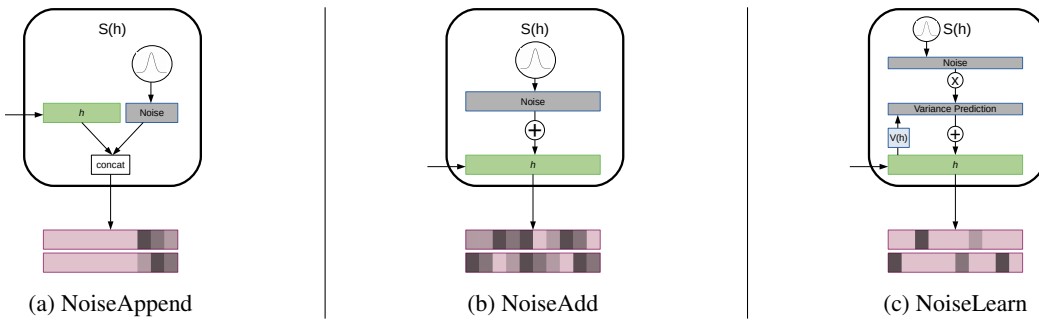

(a) NoiseAppend  (b) NoiseAdd  (c) NoiseLearn

Figure 2: Diagrams of the different approaches to deriving a distribution from the latent shape representation $h$.

Table 1: Detailed description of architectures evaluated. MLP is not present in the 10k family as it is cannot be reduced to that parameter count while still producing 1024 points.

| | Network Architecture | Encoder Hidden | Latent Vector | Appended Noise | $V(h)$ Hidden | Decoder Hidden | Parameter Count |
|---|---|---|---|---|---|---|---|
| **4M** | MLP | 64, 64, 256, 256, 512 | 1024 | | | 1024, 512, 512, 512 | 4433600 |
| | NoiseAppend | 64, 64, 256, 256, 512 | 1024 | 32 | | 2048, 512, 512, 256, 256, 128 | 4468291 |
| | NoiseAdd | 64, 64, 256, 256, 512 | 1024 | | | 2048, 512, 512, 256, 256, 128 | 4402756 |
| | NoiseLearn | 64, 64, 256, 256, 512 | 1024 | | 128, 32 | 2048, 512, 512, 256, 256, 128 | 4573059 |
| **2M** | MLP | 64, 64, 256, 256, 512 | 256 | | | 256, 512, 512, 256 | 1739456 |
| | NoiseAppend | 64, 64, 256, 256, 512 | 256 | 32 | | 256, 512, 512, 256 | 1675843 |
| | NoiseAdd | 64, 64, 256, 256, 512 | 256 | | | 256, 512, 512, 256 | 1643076 |
| | NoiseLearn | 64, 64, 256, 256, 512 | 256 | | 128, 32 | 256, 512, 512, 256 | 1688963 |
| **500k** | MLP | 64, 128, 128 | 64 | | | 256, 256, 128 | 548032 |
| | NoiseAppend | 64, 128, 128 | 64 | 16 | | 512, 512, 256, 128 | 507459 |
| | NoiseAdd | 64, 128, 128 | 64 | | | 512, 512, 256, 128 | 499268 |
| | NoiseLearn | 64, 128, 128 | 64 | | 32 | 512, 512, 256, 128 | 503555 |
| **100k** | MLP | 64, 256 | 64 | | | 32 | 138048 |
| | NoiseAppend | 64, 256 | 64 | 16 | | 256, 128, 64 | 97923 |
| | NoiseAdd | 64, 256 | 64 | | | 256, 128, 64 | 93828 |
| | NoiseLearn | 64, 256 | 64 | | 32 | 256, 128, 64 | 98115 |
| **10k** | NoiseAppend | 64, 32 | 32 | 16 | | 64, 64, 16 | 12595 |
| | NoiseAdd | 64, 32 | 32 | | | 64, 64, 16 | 11572 |
| | NoiseLearn | 64, 32 | 32 | | (none) | 64, 64, 16 | 12659 |

conserving the information necessary to accurately reconstruct it without introducing any additional hyperparameters.

## 4 EVALUATION

We evaluated each decoding architecture by training several instantiations of each architecture on a point cloud auto-encoding problem derived from the ModelNet40 dataset, which consists of over 12,000 3D models of 40 different common object classes. The dataset has a prescribed train/test split, with approximately 9800 models in the training dataset and 2500 in the test dataset. We randomly select 10% of the training data to use for validation during training.

Before training, each object model in the ModelNet40 dataset is used to generate a uniformly-sampled point cloud with 4096 points which is then scaled to fit within the unit sphere. For all auto-encoder network models, at each iteration of training, the point clouds are randomly downsampled to 1024 points before being used to update the network parameters. The helps reduce the computational cost of training and also encouraging better generalization. During training, each decoded point cloud consists of 1024 points.

We use the Chamfer distance as the loss function due to its relative speed and capability to directly compare point clouds of unequal sizes without modification. Each network is trained for 100 epochs using the ADAM optimizer with an initial learning rate of $10^{-3}$, where each epoch performs a parameter update on each training example. The learning rate is decreased by a factor of 10 at epoch 50 and epoch 80. We trained five instantiations of each of the four network architectures with each instantiation varying the number of parameters as shown in Table 1 (note that we were not able to scale down the MLP for the smallest parameter setting). For each instantiation we ran the entire training process 15 times and all results show average performance across the 15 runs.

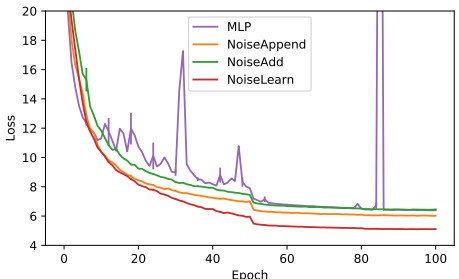

Figure 3: Average validation losses of each architecture in the 2M family during training.

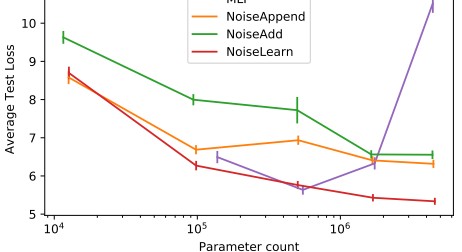

Figure 4: Test performance of each architecture at different parameter counts. Error bars show the 95% confidence interval of the mean.

All code and infrastructure for "push-button" replication of our experiments open-source (Github/Gitlab location removed for anonymous review - code will be privately made available to reviewers through a comment approximately a week after submission).

## 4.1 PERFORMANCE COMPARISON

**Quantitative Results.** Figure 3 shows the validation loss along the learning curves for the 2M parameter instantiation of each architecture. The relative ordering of the architectures is consistent after the initial phase of training, with all curves flattening out by 100 epochs.

First, the large jumps in the MLP training (due to unstable training runs) show that it was much less stable to training compared to the sample-based architectures. While effort was spent trying to specialize the training configuration for the MLP, stability remained an issue.[1] In contrast the runs for each sample based architecture were stable and similar. Ignoring the MLP stability, it performs similarly to worst performing sample-based architectureby the end of the learning curve.

The three sample based architectures show rapid progress early in learning and then settle into a consistent ordering with NoiseLearn performing best, followed by NoiseAppend, and then NoiseAdd. This suggests that NoiseAdd's approach of adding uniform noise to the latent representation may be obscuring information needed for accurate reconstruction, compared to NoiseAppend, which separates noise from the shape representation. On the other hand, we see that while NoiseLearn also adds noise to the latent representation, it is able to outperform NoiseAppend. This indicates the importance of being able to intelligently select how much noise to add to different components of the representation. Apparently, this allows NoiseLearn to avoid obscuring critical information in the latent representation needed for accurate reconstruction.

Figure 4 shows the average test set performance after 100 epochs of training of each size instantiation of the four architectures (note the log scale). The appendix also shows more detailed results broken down for each of 5 selected object classes. The three sample based architectures show relatively consistent improvement in performance as the sizes grow by orders of magnitude. Rather, the MLP shows initial improvement, but then performance decreases significantly past 100K parameters. We looked further into the behavior of the MLP architecture for the larger parameter sets. We observed that the larger MLPs showed a similar decrease in performance on even the training data, indicating that the problem is not necessarily overfitting but also difficulty of the optimization. It is possible that with substantially more epochs the MLP performance would improve, but at great cost.

This indicates that the MLP architecture is much less efficient at exploiting larger network sizes than the more structured sample-based architectures. It is possible that the architecture and training hyperparameters could be tweaked to improve the large MLP networks' performance, such as by adding additional regularization via weight decay or other mechanisms. However, we consider this tweaking to be outside the scope of this work, and note that none of the sampling based architectures required any such tweaking to achieve competitive performance at all parameter counts.

---

[1]We attempted to rectify this by adding batch normalization to layers in the MLP decoder to make it more similar to the architecture of the sampling approaches. While this stabilizied training, it also prevented the MLP architecture from achieving competitive performance, as its loss more than doubled the next best architectures'.

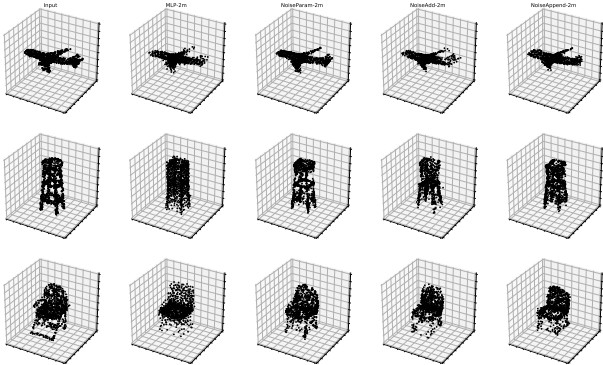

Figure 5: Examples of networks' auto-encoding results on several previously unseen objects.

Overall the results give strong evidence that the sample-based architectures encode a bias that is much better matched to decoding the class of shapes in ModelNet40 compared to MLPs. The structured sample-based architectures, compared to the MLP, result in more stable learning and the ability to continually improve as the architectures grow in size. Further, we see that the NoiseLearn architecture, which avoids the need to specify hyperparameters to control the amount of noise performs the best, or near best, across all network sizes and number of epochs.

**Illustrative Qualitative Results.** Figure 5 shows each network's performance three test objects not seen in the training data. The point cloud decoded by the MLP network appears to be more evenly distributed spatially, while the sampling-based approaches are better able to capture finer detail in the target shape, such as the stool's thin legs and crossbars. Among the sample-based approaches, no single approach is clearly dominant in terms of visual quality across the objects. It is interesting that all of the sample-based architectures tend to miss the same type of object details, e.g. the jets on the plane or the leg cross bars on the chair, which may be due to limitations of the PointNet encoders sized and/or architecture. Nevertheless, it is quite interesting that a single relatively small latent vector representation is able to encode the level of detail exibited in these results.

## 4.2  NOISE EXAMINATION

Each sampling architecture defines a function from the latent shape representation to a point feature distribution. The underlying latent representation inherently defines the manifold of the encoded shape. Rather, the injected noise (either via appending or addition) can be viewed as playing the role of indexing locations on the manifold for the generated point. Effectively, the primary difference between the sample-based architectures is how they use the noise to index locations and traverse the manifolds. Below we aim to better understand this relationship between noise and spatial indexing and how the architectures differ in that respect.

In Figure 6 we demonstrate how each architecture uses noise by controlling the variance introduced to a trained network in two different ways. To examine how the decoder's output is influenced by individual elements of noise we show the output of these networks when all but one of the noise elements is held at a constant near-zero value. In the lower plots, we show the decoder's behavior when it only receives the union of the noise elements above. This demonstrates both how the network learns to exploit individual elements of noise and how the decoder combines those elements to produce a point cloud that spans the entire shape.

For NoiseAppend all of the noise is of equal magnitude, so we just examine the first five elements of noise in its noise vector. NoiseLearn predicts individual variances for each element in the dense shape encoding, enabling us to select the five elements of noise with the highest variance, and therefore presumably the biggest contribution to the decoded point cloud. The appendix contains additional examples of noise manipulation.

The plots shown in Figure 6 give us some insight into how the networks use noise to complete the decoded shape. Each individual element of noise appears to correspond to a learned path along the

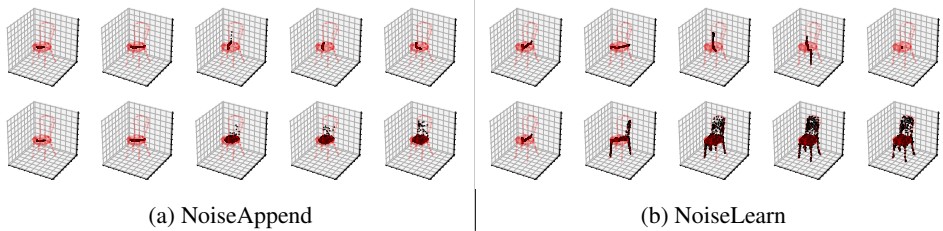

| (a) NoiseAppend | (b) NoiseLearn |

Figure 6: Behavior of NoiseAppend and NoiseLearn when the noise is manipulated. The $i$th figure in the top row shows the decoded point clouds when all but the $i$th variance is set to zero. The same figure in the bottom row shows the decoded point clouds when all but the first $i$ variances are set to zero. The faint red points show the network's input.

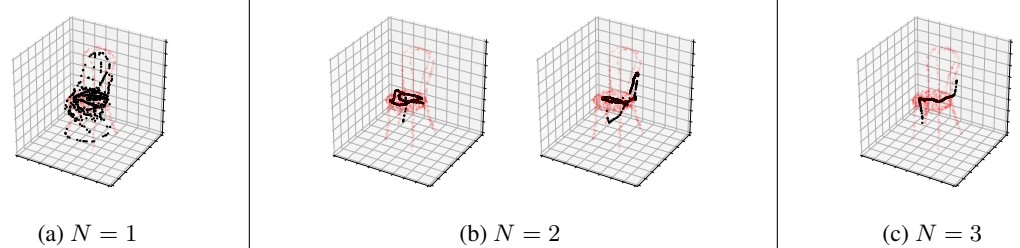

| (a) $N = 1$ | (b) $N = 2$ | (c) $N = 3$ |

Figure 8: Contribution of individual channels of noise when the NoiseAppend architecture is modified to only append $N$ elements of noise to the shape representation.

surface of the learned shape. The final point cloud then seems to be produced by 'extruding' along those paths.

NoiseLearn's use of only four significant elements of noise suggests that in this domain only three or four elements of noise is sufficient to achieve good coverage of the target shape. Figure 8 shows how individual noise channels change when the NoiseAppend architecture is modified to only append one, two, and three noise elements.

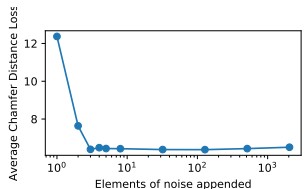

With only one element of noise, we can see that the network effectively has to learn a single path that spans as much of the target shape as possible. With two elements of noise, the network instead seems to learn individual 'loops' around the object which are transformed and rotated as necessary. Once the network has access to three elements of noise, we see the same behavior as the functional networks of learning small paths on the object's surface.

Figure 7: Average final validation loss achieved by NoiseAppend networks when varying the number of elements of noise appended.

If too little noise can seriously hurt NoiseLearn's performance, does adding too much noise do the same? Figure 7 shows the NoiseAppend architecture trained with different amounts of added noise to see if the same performance dropoff is present at both extremes. It appears that even when the noise vector is much larger than the dense shape representation, the decoder's overall performance is not impacted. However, note that adding large amounts of noise does significantly increase the parameter count, so there is a nontrivial cost to doing this.

## 5    CONCLUSION

In this work, we evaluated and compared several realizations of a sample-based point cloud decoder architecture. We show that these sampling approaches are competitive with or outperform the MLP approach while using fewer parameters and providing better functionality. These advantages over the baseline suggest that sample based point cloud decoders should be the default approach when a network needs to produce independent point samples of some underlying function or object. To further this this area of research, we provide a complete open-source implementation of our tools used to train and evaluate these networks.

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

## A  APPENDIX

### A.1  FULL PERFORMANCE TABLES

The tables below show each architecture's average loss on each individual class in the ModelNet40 dataset. The best-performing network is bolded for each object class.

Table 2: Average loss of 2M family architectures (± stddev) on test dataset

|  | airplane | bathtub | bed | bench | bookshelf |
|---|---|---|---|---|---|
| MLP-2m | 3.39 ± 1.89 | 5.67 ± 1.78 | 5.67 ± 1.41 | 5.05 ± 2.73 | 6.42 ± 2.40 |
| NoiseAppend-2m | 3.72 ± 0.88 | 6.10 ± 1.37 | 5.84 ± 1.35 | 5.10 ± 1.54 | 6.74 ± 1.62 |
| NoiseAdd-2m | 4.08 ± 1.10 | 6.23 ± 1.50 | 6.05 ± 1.42 | 5.36 ± 1.72 | 6.90 ± 1.78 |
| NoiseLearn-2m | **2.97 ± 0.73** | **5.31 ± 1.30** | **5.15 ± 1.28** | **3.87 ± 1.27** | **5.95 ± 1.51** |
|  | bottle | bowl | car | chair | cone |
| MLP-2m | **2.69 ± 1.49** | **6.95 ± 3.26** | 5.96 ± 1.28 | 7.04 ± 2.34 | 7.02 ± 49.08 |
| NoiseAppend-2m | 4.13 ± 1.40 | 10.10 ± 4.46 | 6.34 ± 1.08 | 6.78 ± 2.25 | 5.22 ± 2.62 |
| NoiseAdd-2m | 4.03 ± 1.70 | 9.43 ± 4.23 | 6.46 ± 1.15 | 7.29 ± 2.42 | 5.21 ± 2.86 |
| NoiseLearn-2m | 3.35 ± 1.24 | 7.46 ± 3.27 | **5.83 ± 1.04** | **5.15 ± 1.85** | **4.32 ± 2.09** |
|  | cup | curtain | desk | door | dresser |
| MLP-2m | 7.77 ± 2.84 | 2.90 ± 1.55 | 10.78 ± 4.04 | 2.64 ± 1.88 | **5.95 ± 4.77** |
| NoiseAppend-2m | 8.67 ± 2.83 | 3.68 ± 1.04 | 7.53 ± 1.97 | 3.18 ± 0.82 | 7.13 ± 1.74 |
| NoiseAdd-2m | 9.08 ± 2.83 | 3.53 ± 1.24 | 8.14 ± 2.20 | 3.13 ± 1.05 | 7.24 ± 1.77 |
| NoiseLearn-2m | **7.63 ± 2.70** | **2.85 ± 0.88** | **6.00 ± 1.70** | **2.52 ± 0.61** | 6.32 ± 1.48 |
|  | flower_pot | glass_box | guitar | keyboard | lamp |
| MLP-2m | 9.15 ± 4.87 | **5.81 ± 1.70** | 2.65 ± 0.89 | **2.18 ± 0.49** | 13.41 ± 31.22 |
| NoiseAppend-2m | 9.83 ± 3.96 | 7.16 ± 1.79 | 2.17 ± 0.72 | 2.75 ± 0.68 | 8.87 ± 11.19 |
| NoiseAdd-2m | 10.02 ± 3.95 | 7.39 ± 1.94 | 2.05 ± 0.74 | 2.69 ± 0.66 | 8.41 ± 7.15 |
| NoiseLearn-2m | **8.07 ± 3.35** | 6.59 ± 1.63 | **1.45 ± 0.53** | 2.26 ± 0.50 | **6.49 ± 7.05** |
|  | laptop | mantel | monitor | night_stand | person |
| MLP-2m | 3.79 ± 0.65 | 7.29 ± 2.16 | 6.51 ± 3.35 | 7.26 ± 2.43 | 5.55 ± 2.85 |
| NoiseAppend-2m | 4.57 ± 0.85 | 6.65 ± 1.41 | 6.36 ± 1.97 | 7.53 ± 1.96 | 5.21 ± 1.68 |
| NoiseAdd-2m | 4.73 ± 0.98 | 6.78 ± 1.38 | 6.53 ± 2.08 | 7.67 ± 1.94 | 5.08 ± 2.00 |
| NoiseLearn-2m | **3.25 ± 0.50** | **5.59 ± 1.18** | **5.33 ± 1.75** | **6.35 ± 1.80** | **3.87 ± 1.38** |
|  | piano | plant | radio | range_hood | sink |
| MLP-2m | 10.70 ± 2.85 | 10.06 ± 5.18 | 5.73 ± 1.57 | 7.37 ± 2.84 | 9.16 ± 4.50 |
| NoiseAppend-2m | 8.67 ± 2.10 | 9.87 ± 4.82 | 5.94 ± 1.90 | 6.83 ± 1.53 | 6.85 ± 2.09 |
| NoiseAdd-2m | 8.82 ± 2.15 | 10.45 ± 5.21 | 6.06 ± 1.87 | 7.03 ± 1.53 | 7.05 ± 2.18 |
| NoiseLearn-2m | **7.22 ± 1.77** | **8.38 ± 4.20** | **5.12 ± 1.83** | **5.76 ± 1.25** | **5.77 ± 1.77** |
|  | sofa | stairs | stool | table | tent |
| MLP-2m | 6.47 ± 1.73 | 11.44 ± 6.03 | 7.00 ± 3.15 | 4.56 ± 1.90 | 7.42 ± 2.47 |
| NoiseAppend-2m | 6.60 ± 1.62 | 8.16 ± 3.29 | 6.08 ± 2.35 | 4.19 ± 1.32 | 7.17 ± 1.74 |
| NoiseAdd-2m | 6.79 ± 1.68 | 8.37 ± 3.52 | 6.78 ± 2.85 | 4.48 ± 1.59 | 7.16 ± 1.84 |
| NoiseLearn-2m | **5.84 ± 1.39** | **6.18 ± 2.46** | **4.75 ± 1.71** | **3.15 ± 1.08** | **6.05 ± 1.72** |
|  | toilet | tv_stand | vase | wardrobe | xbox |
| MLP-2m | 7.92 ± 2.57 | 6.92 ± 2.36 | **5.92 ± 3.49** | **5.53 ± 2.49** | 6.12 ± 3.03 |
| NoiseAppend-2m | 8.21 ± 1.81 | 6.79 ± 1.78 | 7.60 ± 3.75 | 6.24 ± 1.24 | 6.59 ± 1.46 |
| NoiseAdd-2m | 8.51 ± 1.89 | 7.01 ± 1.84 | 7.73 ± 4.62 | 6.42 ± 1.38 | 6.67 ± 1.47 |
| NoiseLearn-2m | **6.83 ± 1.54** | **6.01 ± 1.66** | 6.25 ± 2.97 | 5.54 ± 1.13 | **5.75 ± 0.92** |

Table 3: Average loss of 100k family architectures (± stddev) on test dataset

|  | airplane | bathtub | bed | bench | bookshelf |
|---|---|---|---|---|---|
| MLP-100k | 4.53 ± 1.27 | 5.88 ± 1.59 | 5.76 ± 1.20 | 5.52 ± 1.57 | **6.37 ± 2.06** |
| NoiseAppend-100k | 4.76 ± 1.10 | 6.16 ± 1.50 | 6.07 ± 1.37 | 5.35 ± 1.78 | 7.00 ± 2.01 |
| NoiseAdd-100k | 5.98 ± 1.48 | 7.49 ± 1.61 | 7.07 ± 1.40 | 6.66 ± 2.20 | 8.06 ± 2.33 |
| NoiseLearn-100k | **4.16 ± 1.06** | **5.77 ± 1.49** | **5.75 ± 1.36** | **4.91 ± 1.76** | 6.62 ± 1.94 |
|  | bottle | bowl | car | chair | cone |
| MLP-100k | **2.79 ± 1.45** | **8.17 ± 3.40** | **5.61 ± 1.02** | 6.97 ± 2.12 | **4.74 ± 2.92** |
| NoiseAppend-100k | 3.70 ± 1.57 | 8.95 ± 4.15 | 6.39 ± 1.14 | 7.07 ± 2.22 | 5.40 ± 3.13 |
| NoiseAdd-100k | 4.81 ± 2.12 | 11.56 ± 4.63 | 7.47 ± 1.41 | 8.27 ± 2.43 | 7.11 ± 4.11 |
| NoiseLearn-100k | 3.35 ± 1.44 | 8.32 ± 3.80 | 6.11 ± 1.08 | **6.70 ± 2.20** | 4.89 ± 2.87 |
|  | cup | curtain | desk | door | dresser |
| MLP-100k | **7.48 ± 2.60** | **2.99 ± 1.06** | 9.58 ± 2.86 | 2.82 ± 1.38 | **5.75 ± 1.52** |
| NoiseAppend-100k | 9.03 ± 2.91 | 3.33 ± 1.11 | 8.74 ± 2.47 | 2.96 ± 0.81 | 7.19 ± 1.63 |
| NoiseAdd-100k | 10.80 ± 2.82 | 4.02 ± 1.30 | 9.77 ± 2.54 | 3.82 ± 1.16 | 9.18 ± 1.92 |
| NoiseLearn-100k | 8.46 ± 2.87 | 3.12 ± 1.14 | **8.20 ± 2.45** | **2.67 ± 0.71** | 6.71 ± 1.62 |
|  | flower_pot | glass_box | guitar | keyboard | lamp |
| MLP-100k | **9.10 ± 4.44** | **6.30 ± 1.53** | 2.05 ± 0.78 | 2.55 ± 0.55 | 15.12 ± 27.38 |
| NoiseAppend-100k | 9.65 ± 3.99 | 7.43 ± 1.87 | 1.85 ± 0.70 | 2.57 ± 0.62 | **10.69 ± 13.88** |
| NoiseAdd-100k | 11.61 ± 4.72 | 9.37 ± 2.54 | 2.36 ± 0.92 | 3.23 ± 0.84 | 12.89 ± 13.45 |
| NoiseLearn-100k | 9.27 ± 4.07 | 6.99 ± 1.75 | **1.72 ± 0.68** | **2.41 ± 0.56** | 10.14 ± 14.71 |
|  | laptop | mantel | monitor | night_stand | person |
| MLP-100k | 5.10 ± 0.83 | 7.29 ± 1.86 | **5.87 ± 1.92** | **6.86 ± 2.13** | 4.98 ± 2.16 |
| NoiseAppend-100k | 4.16 ± 0.51 | 7.00 ± 1.39 | 6.48 ± 2.06 | 7.68 ± 2.03 | 5.12 ± 2.02 |
| NoiseAdd-100k | 5.02 ± 0.67 | 8.54 ± 1.84 | 7.61 ± 2.43 | 9.37 ± 2.37 | 6.14 ± 2.40 |
| NoiseLearn-100k | **3.74 ± 0.51** | **6.45 ± 1.40** | 6.09 ± 1.98 | 7.20 ± 1.92 | **4.79 ± 1.95** |
|  | piano | plant | radio | range_hood | sink |
| MLP-100k | **8.79 ± 2.19** | 10.79 ± 5.86 | **5.14 ± 1.50** | 6.99 ± 1.82 | 9.18 ± 4.13 |
| NoiseAppend-100k | 9.47 ± 2.36 | 10.25 ± 5.30 | 5.91 ± 2.11 | 7.24 ± 1.52 | 8.06 ± 2.58 |
| NoiseAdd-100k | 11.04 ± 2.81 | 11.46 ± 5.99 | 7.70 ± 2.85 | 8.71 ± 1.82 | 10.02 ± 3.21 |
| NoiseLearn-100k | 8.91 ± 2.25 | **9.98 ± 5.21** | 5.52 ± 1.90 | **6.87 ± 1.48** | **7.45 ± 2.44** |
|  | sofa | stairs | stool | table | tent |
| MLP-100k | **6.17 ± 1.34** | 9.22 ± 4.20 | 7.25 ± 2.92 | 5.51 ± 2.01 | **6.25 ± 1.58** |
| NoiseAppend-100k | 6.72 ± 1.59 | 8.74 ± 3.71 | 6.78 ± 2.69 | 4.90 ± 1.80 | 7.19 ± 1.92 |
| NoiseAdd-100k | 7.68 ± 1.96 | 10.57 ± 4.33 | 8.56 ± 3.57 | 6.10 ± 1.96 | 8.93 ± 2.61 |
| NoiseLearn-100k | 6.43 ± 1.51 | **8.08 ± 3.40** | **6.30 ± 2.45** | **4.31 ± 1.66** | 6.76 ± 1.83 |
|  | toilet | tv_stand | vase | wardrobe | xbox |
| MLP-100k | 8.50 ± 1.59 | 6.84 ± 2.02 | **5.99 ± 3.33** | **5.43 ± 1.77** | **5.59 ± 1.52** |
| NoiseAppend-100k | 8.89 ± 1.73 | 7.14 ± 1.96 | 7.39 ± 3.55 | 6.54 ± 1.33 | 6.81 ± 1.73 |
| NoiseAdd-100k | 10.84 ± 2.23 | 8.31 ± 2.26 | 9.40 ± 4.26 | 8.43 ± 1.61 | 8.66 ± 1.82 |
| NoiseLearn-100k | **8.34 ± 1.71** | **6.82 ± 1.87** | 6.82 ± 3.41 | 6.04 ± 1.30 | 6.32 ± 1.24 |

Table 4: Average loss of 4M family architectures ($\pm$ stddev) on test dataset

|  | airplane | bathtub | bed | bench | bookshelf |
|---|---|---|---|---|---|
| MLP-4m | 8.45 ± 2.38 | 10.04 ± 2.81 | 9.42 ± 2.72 | 10.31 ± 4.79 | 9.19 ± 3.94 |
| NoiseAppend-4m | 3.68 ± 0.90 | 6.06 ± 1.42 | 5.84 ± 1.37 | 5.14 ± 1.45 | 6.63 ± 1.59 |
| NoiseAdd-4m | 4.07 ± 1.11 | 6.27 ± 1.53 | 6.11 ± 1.43 | 5.58 ± 1.79 | 6.89 ± 1.79 |
| NoiseLearn-4m | **2.95 ± 0.72** | **5.30 ± 1.29** | **5.11 ± 1.26** | **3.81 ± 1.26** | **5.82 ± 1.54** |
|  | bottle | bowl | car | chair | cone |
| MLP-4m | 4.59 ± 2.58 | 14.06 ± 4.90 | 9.01 ± 2.77 | 13.51 ± 5.09 | 9.45 ± 6.15 |
| NoiseAppend-4m | 3.93 ± 1.40 | 9.61 ± 3.89 | 6.39 ± 1.12 | 6.62 ± 2.18 | 5.32 ± 2.70 |
| NoiseAdd-4m | 4.03 ± 1.62 | 9.49 ± 4.40 | 6.44 ± 1.17 | 7.39 ± 2.46 | 5.21 ± 2.56 |
| NoiseLearn-4m | **3.22 ± 1.19** | **7.39 ± 3.09** | **5.86 ± 1.04** | **5.08 ± 1.82** | **4.28 ± 2.02** |
|  | cup | curtain | desk | door | dresser |
| MLP-4m | 12.07 ± 3.68 | 5.65 ± 5.26 | 15.68 ± 6.05 | 5.09 ± 4.61 | 10.15 ± 3.54 |
| NoiseAppend-4m | 8.58 ± 2.93 | 3.72 ± 1.01 | 7.64 ± 2.00 | 3.18 ± 0.81 | 6.96 ± 1.68 |
| NoiseAdd-4m | 9.21 ± 2.85 | 3.64 ± 1.26 | 8.35 ± 2.27 | 3.06 ± 0.82 | 7.21 ± 1.80 |
| NoiseLearn-4m | **7.50 ± 2.68** | **2.81 ± 0.89** | **5.90 ± 1.64** | **2.49 ± 0.63** | **6.27 ± 1.48** |
|  | flower_pot | glass_box | guitar | keyboard | lamp |
| MLP-4m | 12.45 ± 5.96 | 10.98 ± 3.73 | 5.44 ± 2.36 | 4.24 ± 1.74 | 24.83 ± 42.31 |
| NoiseAppend-4m | 9.38 ± 3.53 | 7.07 ± 1.79 | 2.19 ± 0.69 | 2.74 ± 0.62 | 9.36 ± 12.32 |
| NoiseAdd-4m | 10.15 ± 3.88 | 7.56 ± 2.10 | 2.11 ± 0.77 | 2.69 ± 0.63 | 8.15 ± 5.93 |
| NoiseLearn-4m | **7.87 ± 3.23** | **6.53 ± 1.61** | **1.45 ± 0.51** | **2.21 ± 0.48** | **6.28 ± 6.67** |
|  | laptop | mantel | monitor | night_stand | person |
| MLP-4m | 13.72 ± 5.49 | 13.97 ± 5.53 | 11.00 ± 6.65 | 12.64 ± 4.27 | 7.30 ± 3.16 |
| NoiseAppend-4m | 4.62 ± 0.96 | 6.64 ± 1.40 | 6.29 ± 2.00 | 7.36 ± 1.93 | 5.06 ± 1.50 |
| NoiseAdd-4m | 5.20 ± 1.26 | 6.83 ± 1.44 | 6.64 ± 2.12 | 7.65 ± 2.02 | 5.37 ± 1.96 |
| NoiseLearn-4m | **3.27 ± 0.53** | **5.57 ± 1.18** | **5.19 ± 1.72** | **6.24 ± 1.77** | **3.74 ± 1.31** |
|  | piano | plant | radio | range_hood | sink |
| MLP-4m | 14.19 ± 4.32 | 13.97 ± 7.85 | 10.28 ± 3.85 | 13.17 ± 4.07 | 14.79 ± 5.38 |
| NoiseAppend-4m | 8.49 ± 2.00 | 9.77 ± 4.76 | 5.85 ± 1.81 | 6.68 ± 1.43 | 6.70 ± 2.11 |
| NoiseAdd-4m | 8.93 ± 2.18 | 10.68 ± 5.26 | 6.09 ± 1.99 | 7.01 ± 1.57 | 7.03 ± 2.17 |
| NoiseLearn-4m | **7.09 ± 1.74** | **8.17 ± 4.09** | **5.06 ± 1.83** | **5.75 ± 1.23** | **5.67 ± 1.84** |
|  | sofa | stairs | stool | table | tent |
| MLP-4m | 10.07 ± 3.23 | 19.81 ± 8.58 | 12.93 ± 4.80 | 10.41 ± 4.93 | 11.56 ± 3.88 |
| NoiseAppend-4m | 6.56 ± 1.59 | 8.11 ± 3.08 | 6.00 ± 2.33 | 4.22 ± 1.35 | 7.03 ± 1.74 |
| NoiseAdd-4m | 6.81 ± 1.72 | 8.60 ± 3.54 | 6.94 ± 3.35 | 4.71 ± 1.64 | 7.24 ± 1.99 |
| NoiseLearn-4m | **5.77 ± 1.37** | **6.20 ± 2.42** | **4.46 ± 1.37** | **3.16 ± 1.08** | **5.97 ± 1.58** |
|  | toilet | tv_stand | vase | wardrobe | xbox |
| MLP-4m | 16.68 ± 5.08 | 10.99 ± 4.61 | 10.03 ± 6.06 | 9.21 ± 4.15 | 9.51 ± 5.13 |
| NoiseAppend-4m | 7.97 ± 1.73 | 6.79 ± 1.74 | 7.28 ± 3.46 | 6.19 ± 1.21 | 6.35 ± 1.34 |
| NoiseAdd-4m | 8.62 ± 1.88 | 7.08 ± 1.94 | 8.33 ± 21.51 | 6.42 ± 1.37 | 6.58 ± 1.50 |
| NoiseLearn-4m | **6.68 ± 1.49** | **5.94 ± 1.63** | **6.15 ± 2.90** | **5.45 ± 1.16** | **5.64 ± 0.88** |

## A.2   NOISE MANIPULATIONS

Below are more examples of noise manipulation in an autoencoder setting in the style of Figure 6.

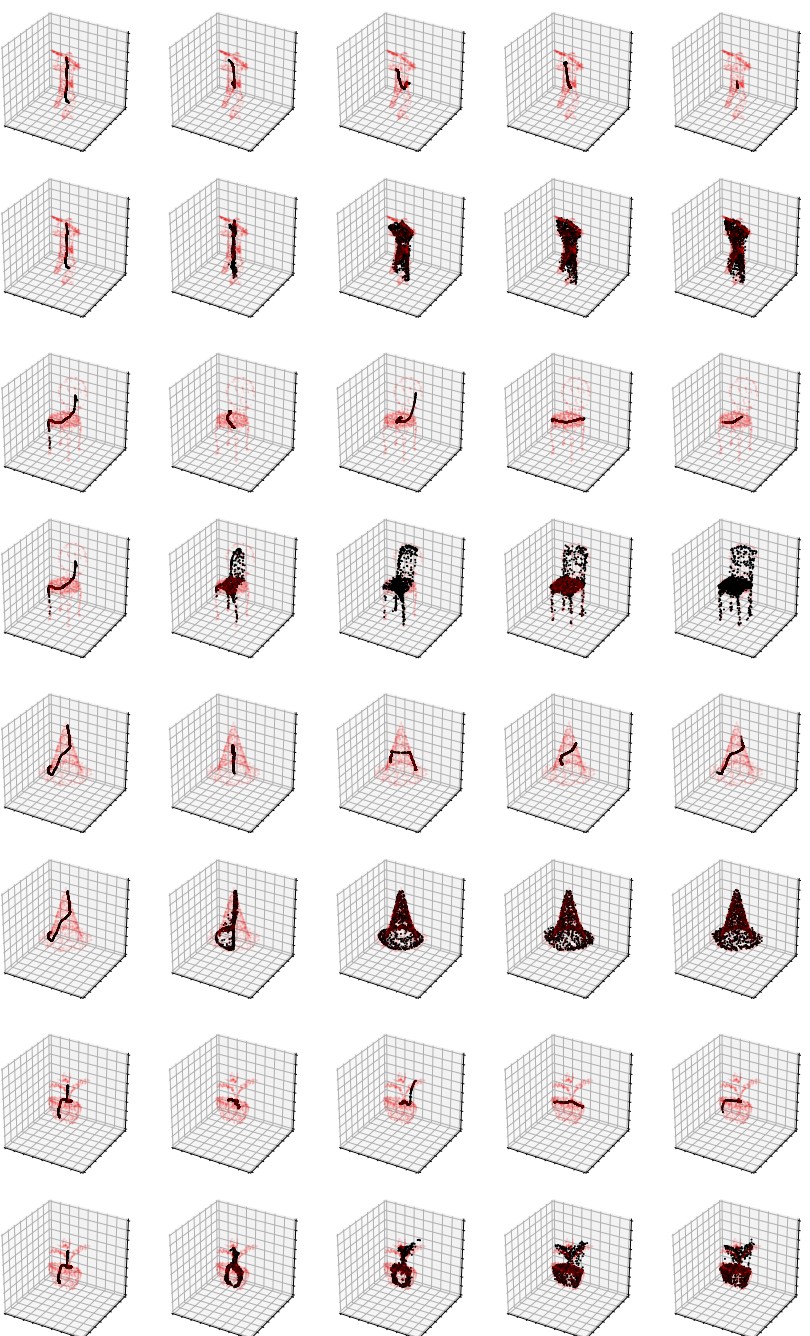

