# OpenReview forum: "Sample-Based Point Cloud Decoder Networks"
_ICLR.cc/2020/Conference — Reject_

### Official Review · AnonReviewer2 · 2019-10-21
**Official Blind Review #2**

**Rating:** 3

**Review:**

The paper is about decoder networks for 3D point cloud data, i.e., given a latent vector that encodes shape information, the decoder network outputs a 3D point cloud. Previous approaches like PU-Net, or TopNet have used an MLP with a fixed number of output points for this task. The approach in this paper takes inspiration from the work of Li et al. and extends a sample based method for the decoder network. Specifically, the core contribution is to sample from a Gaussian distribution with the latent vector as mean and a variance computed by a small network conditioned on the latent vector. This sampled vector is then feed to an MLP to generate a 3D point estimate. The sampling approach is compared to the method of Li et al. (concatenating noise values to the latent value), and to directly adding Gaussian noise to the latent vector on auto-encoding shapes from the ModelNet40 dataset. Further studies on the impact of the noise on the reconstructed shapes are presented.

I argue for a weak reject of this paper (in its current form) because (1) the presented method seems to be too incremental, and (2) I am missing experiments that compare to other state-of-the-art methods (e.g. on the task of 3D reconstruction from single images).

Regarding the first point, let's start with the related work that is missing some very relevant paper. I would like to see the relation and comparison to implicit surface models (e.g., Park et al. "DeepSDF: Learning Continuous Signed Distance Functions for Shape Representation", or Mescheder et al. "Occupancy Networks: Learning 3D Reconstruction in Function Space") that sample not based on a random vector, but on the 3D position itself. Further, I see Fan et al. "A point set generation network for 3d object reconstruction from a single image" in the references, but not where it is discussed in the paper. Further, Yang et al. "FoldingNet: Point Cloud Auto-encoder via Deep Grid Deformation" and Groueix et al. "AtlasNet: A Papier-Mâché Approach to Learning 3D Surface Generation" seem to be quite relevant.
Further, the core contribution is to add a noise vector to the latent vector, instead of appending it to the latent vector as in Li et al. The presented evaluations in Fig. 3 and Fig. 4 show that this might be slightly beneficial, but I am not convinced how this performance increase would translate to a other tasks.

This is also my second major point of criticism: The presented evaluations are nice, but also seem like a toy problem. What would have been interesting is to compare the presented decoder network to state-of-the-art methods on other tasks presented in the literature. One example might be 3D reconstruction from a single image (cf. Fan et al.), or shape completion (cf. Park et al.). Even for the auto-encoding setting it would be possible to compare to other methods (cf. Yang et al., PointFlow).

One part that might lower the performance on the above mentioned task is the inferior encoder network (PointNet) that takes no neighbourhood information into account. There are quite a few stronger methods that can be considered, e.g. PointNet++, or any of the recently proposed point convolution methods.

In Fig. 3 and 4 the loss seems to be quite large, given that a Chamfer distance is used and the shapes are scaled to fit into a unit sphere. What function is used for dist? And are there severe outliers in the generated point clouds?

On page 5 it is stated that the Chamfer distance is used due to its relative speed. Relative to what?

On page 7 it is stated that the MLP results appear to be more evenly distributed. The last row in Fig. 5 suggests otherwise on the back of the chair. Also, the NoiseAdd and NoiseAppend seem to miss quite some details in the second row (or are just more noisy in general), which one is the proposed NoiseLearn?

Things to improve the paper that dis not impact the score:
- On the second pager, first line, there is an inconsistency in the citing style (use of parentheses)
- What is meant with a "parameterized ReLU" on page 4?
- The point cloud visualizations can be greatly improved. The grid is not necessary and the images are very small, making it hard to see details.

**Experience Assessment:**

I have published one or two papers in this area.

**Review Assessment: Checking Correctness Of Derivations And Theory:**

N/A

**Review Assessment: Checking Correctness Of Experiments:**

I carefully checked the experiments.

**Review Assessment: Thoroughness In Paper Reading:**

I read the paper thoroughly.

---

### Official Review · AnonReviewer4 · 2019-11-03
**Official Blind Review #4**

**Rating:** 3

**Review:**

The paper deals with a task of learning latent representations for 3D point clouds by introducing a series of particular point cloud decoders. The advantage of the proposed architectures is their ability of producing variable-size output 3D point cloud. While the work on point-based architectures is relevant to the current agenda in the vision community, the novelty of this paper is somewhat limited, as the proposed architecture is constrained to one class (PointNets) only, while the decoding network could just as easily be used with other encoders (e.g., DGCNN [1]).


Major comments:

One major shortcoming of the paper is its very limited experimental evaluation, that does not really allow to judge whether the proposed decoder architecture brings the real utility for most problems. For instance, authors only provide results for two distinct experiments: building an auto-encoder and understanding how each architecture uses noise. Much more tasks, such as shape denoising, upsampling, completion, or retrieval could have been explored, too. Only one (simple) dataset is used, i.e. ModelNet, whereas more complex datasets (e.g., ShapeNet, ABC, or datasets related to human shapes) could have been considered.

I wonder why authors didn't compare to (or even cite) [1], which seems to be one of the earliest papers in the area.

Other possible research questions are as follows. What is the effect of the proposed decoder networks on the learned latent representations in the entire architecture? Will the interpolatory and semantic properties of the learned latent spaces be preserved when using the autoencoders?

Having said that, I believe the paper cannot be accepted in the present form, however, I would change my decision, should more experiments validating the architecture be provided.


Minor comments:

* in definition of a point cloud, both n and N are used to indicate its size


[1] Wang, Y., Sun, Y., Liu, Z., Sarma, S. E., Bronstein, M. M., & Solomon, J. M. (2019). Dynamic graph cnn for learning on point clouds. ACM Transactions on Graphics (TOG), 38(5), 146.
[2] Achlioptas, P., Diamanti, O., Mitliagkas, I., & Guibas, L. (2018, July). Learning Representations and Generative Models for 3D Point Clouds. In International Conference on Machine Learning (pp. 40-49).

**Experience Assessment:**

I have read many papers in this area.

**Review Assessment: Checking Correctness Of Derivations And Theory:**

I assessed the sensibility of the derivations and theory.

**Review Assessment: Checking Correctness Of Experiments:**

I assessed the sensibility of the experiments.

**Review Assessment: Thoroughness In Paper Reading:**

I made a quick assessment of this paper.

---

### Official Review · AnonReviewer5 · 2019-11-03
**Official Blind Review #5**

**Rating:** 1

**Review:**


Summary

This paper studies the problem of 3D point cloud autoencoding in a deep learning setup, and in particular, the choice of the architecture of a 3D point cloud decoder. The main claim of the paper is the adaptation of the PointNet architecture (a set of fully-connected layers applied per point) from encoding to decoding purposes. To achieve that, the authors propose three similar mechanisms to inject noise in the bottleneck shape representation, which are used to sample per point feature vectors suitable for a PointNet-like decoder. These include appending a low dimensional gaussian noise sample to the bottleneck shape representation, adding individual gaussian noise sample to each bottleneck shape representation dimension or using trainable variances per bottleneck shape representation dimension. Resulting networks are optimized by minimizing Chamfer distance between the ground truth and output point clouds. The main motivation of their choice of these architectures is their ability to output an arbitrary size point clouds, compared to naive multilayered fully-connected network applied to a bottleneck shape representation and outputting a "flattened" fixed-size point cloud.

To verify their claims, the authors considered autoencoding task for ModelNet40 dataset. They compared a naive multilayered fully-connected decoder to PointNet-like decoders with all three proposed noise-injection mechanisms. This comparison was also performed for 5 variations of the proposed models differing is network size. To compare quantitatively, the authors present all results graphically, showing the dependency of resulting average Chamfer distance from the network size. This graph shows that using trainable per dimension variance for noise injection works consistently better, than other options; also, it shows inconsistent results for the baseline multilayered fully-connected decoder. In qualitative comparisons, the authors compare output point clouds visually and study the influence of sampling from various noise dimensions on the reconstructed points.


Review

Although the authors of this paper touch a very interesting topic in generative modeling of point clouds, I do not believe this paper should be published in its current state.

Cons:
1. The main issue in this paper is the absence of comparisons to any external approaches. Even though comparison to their variation of the baseline model is provided, it is impossible to verify where their results stand compared to the related work. There is a whole stream of works in the same task using ShapeNet dataset, which the authors completely omit. Even if the authors do not want to perform experiments on ShapeNet, they, at least, could have provided comparison to AAE and PC-GAN results from PC-GAN paper [1] for generative modeling metrics on ModelNet40. As the authors stated, there are at least two works, PC-GAN and PointFlow, which allow sampling of the arbitrary size point clouds and comparison to these approaches is crucial (especially since the proposed approach is technically significantly simpler compared to hierarchical GANs from PC-GAN and continuous normalizing flows combined with VAEs from PointFlow papers).

2. The paper misses numerous citations to related work in generative 3D shape modeling:
- voxel grid based, (since the authors mention them): [2, 3, 4, 5, 6, 7]
- point cloud based: [8, 9, 10]
- graph based: [11, 12]
All these approaches propose various types of decoders for 3D shapes represented accordingly and some of them perform evaluations for pure generative, autoencoding task, which, again, can be used for comparison.

3. I believe, there is a factual mistake and arguable diminishing of the current state of the art in the related work section:
The authors state, that both PC-GAN and PointFlow models "inhabit GAN settings". However, PointFlow is, in fact, an approximate likelihood model and is considered as an instance of Variational Autoencoder with a shape prior and a point cloud decoder implemented as continuous normalizing flows (a special case of invertible neural networks). The authors also state, that it is "unclear... if they are capable of outperforming the baseline MLP approach" when they talk about these papers, while, in fact, both PC-GAN and PointFlow explicitly show that by comparing to [9], which use the multilayered fully-connected baseline decoder. It is also not unclear if these approaches need specific loss functions, contrary to what authors say. They were designed so they need their particular loss functions, which does not disqualify them from the comparison. These papers are the main competitors of this work and should be addressed properly instead of being diminished.

Pros:
1. Besides the cons, the paper is written clearly and is easy to follow.

2. Rigorousness with respect to comparison of the proposed options (Table 1 and the fact that every model was retrained 15 times to obtain average result).

3. Qualitative experiments with control over noise dimensions are novel and interesting.

Overall, this could have been a paper about a nice and simple alternative to PC-GAN and PointFlow for arbitrary size point cloud modeling, but, unfortunately, it missed the opportunity to properly address the current state-of-the-art in its current version.


Small remarks and questions for authors:

1. Citation of Valsesia at al. in the related work should be reorganized to be consistent with other citations. Omitting full names there and in case of PC-GAN will also be more consistent.

2. All figures in the paper should be updated for better visibility of both graphics and text. Figures 1-4, 7 - text is barely visible, the scheme blocks and lines are too thin and overall lack readability. Figures 5-6, 8 - point clouds are almost invisible even with zooming, remove coordinate grid in the background, zoom in on the objects, so they take as much space in the images as possible, use bigger point sizes for better visibility.

3. Maybe I do not follow something, why have you ignored applications of your point cloud decoders, and particularly generative modeling task and its evaluation protocols, presented for example in [9]?


[1] C.-L. Li, M. Zaheer, Y. Zhang, B. Póczos and R. Salakhutdinov. Point Cloud GAN. In ICLR Workshop on Deep Generative Models for Highly Structured Data, 2019
[2] C. Choy, D. Xu, J.-Y. Gwak, K. Chen, and S. Savarese. 3D-R2N2: A unified approach for single and multi-view 3d object reconstruction. In ECCV, 2016.
[3] R. Girdhar, D. Fouhey, M. Rodriguez, and A. Gupta. Learning a predictable and gen- erative vector representation for objects. In ECCV, 2016.
[4] J. Wu, C. Zhang, T. Xue, W. Freeman, and J. Tenenbaum. Learning a probabilistic latent space of object shapes via 3D generative-adversarial modeling. In NeurIPS, 2016.
[5] M. Tatarchenko, A. Dosovitskiy, and T. Brox. Octree generating networks: Efficient
convolutional architectures for high-resolution 3D outputs. In ICCV, 2017.
[6] S. R. Richter and S. Roth. Matryoshka Networks: Predicting 3d geometry via nested shape layers. In CVPR, 2018.
[7] R. Klokov, J. Verbeek, and E. Boyer. Probabilistic Reconstruction Networks for 3D Shape Inference from a Single Image. In BMVC, 2019
[8] T. Groueix, M. Fisher, V. Kim, B. Russell, and M. Aubry. AtlasNet: A papier-mâché approach to learning 3D surface generation. In CVPR, 2018.
[9] P. Achlioptas, O. Diamanti, I. Mitliagkas, and L. Guibas. Learning Representations and Generative Models for 3D Point Clouds. In ICML, 2018.
[10] P. Mandikal, K. Navaneet, M. Agarwal, and R. Babu. 3D-LMNet: Latent embedding matching for accurate and diverse 3D point cloud reconstruction from a single image. In BMVC, 2018.
[11] N. Wang, Y. Zhang, Z. Li, Y. Fu, W. Liu, and Y. Jiang. Pixel2Mesh: Generating 3D
mesh models from single RGB images. In ECCV, 2018.
[12] C. Wen, Y. Zhang, Z. Li, and Y. Fu. Pixel2Mesh++: Multi-View 3D Mesh Generation via Deformation. In ICCV, 2019.

**Experience Assessment:**

I have published one or two papers in this area.

**Review Assessment: Checking Correctness Of Derivations And Theory:**

I carefully checked the derivations and theory.

**Review Assessment: Checking Correctness Of Experiments:**

I carefully checked the experiments.

**Review Assessment: Thoroughness In Paper Reading:**

I read the paper at least twice and used my best judgement in assessing the paper.

---

### Decision · Program_Chairs · 2019-12-19

**Decision:**

Reject

**Comment:**

The reviewers have raised several important concerns about the paper that the authors decided not to address.